Conspicuous carotenoid-based pelvic spine ornament in three-spined stickleback populations—occurrence and inheritance

Amundsen CR 1
Nordeide JT 1 Jarle.Nordeide@uin.no
Gjøen HM 2
Larsen B 3
Egeland ES 1
1 Faculty of Biosciences and Aquaculture, University of Nordland , Bodø , Norway
2 Animal and Aquacultural Sciences, Norwegian University of Life Sciences , Ås , Norway
3 Bodø Graduate School of Business, University of Nordland , Bodø , Norway
Hedrick Ann
Electronic publication date: 2015 Apr 2
Publication date: 2015
Volume: 3
Electronic Location ID: e872
Received 2015 Jan 13; Accepted 2015 Mar 11
Copyright: © 2015 Amundsen et al.
Copyright year: 2015
Copyright holder: Amundsen et al.
License: This is an open access article distributed under the terms of the Creative Commons Attribution License, which permits unrestricted use, distribution, reproduction and adaptation in any medium and for any purpose provided that it is properly attributed. For attribution, the original author(s), title, publication source (PeerJ) and either DOI or URL of the article must be cited.
License URL: https://creativecommons.org/licenses/by/4.0/

Keywords: Gasterosteus aculeatus, Stickleback, Ornament, Carotenoid, Pelvic spine, Signal, Female ornament

Funding: University of Nordland This work was funded by internal resources at the University of Nordland. The funders had no role in study design, data collection and analysis, decision to publish, or preparation of the manuscript.

==============================
Reports on reddish carotenoid-based ornaments in female three-spined sticklebacks (Gasterosteus aculeatus) are few, despite the large interest in the species’ behaviour, ornamentation, morphology and evolution. We sampled sticklebacks from 17 sites in north-western Europe in this first extensive study on the occurrence of carotenoid-based female pelvic spines and throat ornaments. The field results showed that females, and males, with reddish spines were found in all 17 populations. Specimens of both sexes with conspicuous red spines were found in several of the sites. The pelvic spines of males were more intensely red compared to the females’ spines, and large specimens were more red than small ones. Fish infected with the tapeworm (Schistocephalus solidus) had drabber spines than uninfected fish. Both sexes had red spines both during and after the spawning period, but the intensity of the red colour was more exaggerated during the spawning period. As opposed to pelvic spines, no sign of red colour at the throat was observed in any female from any of the 17 populations. A rearing experiment was carried out to estimate a potential genetic component of the pelvic spine ornament by artificial crossing and rearing of 15 family groups during a 12 months period. The results indicated that the genetic component of the red colour at the spines was low or close to zero. Although reddish pelvic spines seem common in populations of stickleback, the potential adaptive function of the reddish pelvic spines remains largely unexplained.

Introduction

Sexual selection has dominated the study of behavioural ecology the last 25 years (Andersson, 1994; Milinski, 2014; Simmons, 2014). Although the main focus has been on female choice and males’ elaborate ornaments, the evolution of female ornaments has received attention as well (Amundsen, 2000; Clutton-Brock, 2009; Kraaijeveld, Kraaijeveld-Smit & Komdeur, 2007). Female ornaments come in different varieties and may be female-specific or mutual for both sexes. They may be relatively static over longer time-periods or highly dynamic, for example as signals of fertility or ovulation (e.g., McLennan, 1994; McLennan, 1995; Rowland, Baube & Horan, 1991; Amundsen & Forsgren, 2001). Authors have hypothesized that ornaments signal genetic quality of females (e.g., Zahavi, 1975), or direct benefits for offspring such as non-genetic maternal resources (e.g., Blount, Houston & Møller, 2000; Massaro, Davis & Darby, 2003; Gladbach et al., 2010). However, the resources allocated to ornamentation may also lead to reduced resources available for offspring and thus constrain the females’ investment in offspring (Fitzpatrick, Berglund & Rosenqvist, 1995; Price, 1996; LeBas, Hockham & Ritchie, 2003; Chenoweth, Doughty & Kokko, 2006), giving dishonest female signals (Funk & Tallamy, 2000; Bonduriansky, 2001).

Several hypotheses have been proposed to explain the evolution of female ornaments in mutually ornamented species. The “direct selection hypothesis” suggests that female ornaments are under direct sexual selection by males, or under selection due to competition among females (Amundsen, 2000; Kraaijeveld, Kraaijeveld-Smit & Komdeur, 2007). Thus, according to this hypothesis, female ornaments are honest signals of some aspects of individual quality. The alternative “genetic correlation hypothesis”, predicts that female ornamentation is a genetically correlated response to selection for male ornamentation and this received some support already from Darwin (1871). Later, Lande (1980) suggested that female ornamentation in mutually ornamented species may be just a temporal stage in the evolution of male ornaments. The evolutionary explanations for females’ ornaments remain controversial (Nordeide et al., 2013).

The three-spined stickleback (Gasterosteus aculeatus) has been studied for decades to address diverse topics within ecology, morphology, and evolutionary biology (reviews by (Wootton, 1976; Wootton, 1984; Bell & Foster, 1994; Őstlund-Nilsson, Mayer & Huntingford, 2007)). Male sticklebacks develop the nuptial blue eyes and yellow—reddish carotenoid-based throat (reviewed by Rowland, 1994) (for simplicity, we refer to the yellow—reddish carotenoid-based ornaments as “red” in the rest of this paper). Red serves as a strong signal eliciting territorial aggression (ter Pelkwijk & Tinbergen, 1937; Tinbergen, 1948) or a dual effect of aggression and fear in male three-spined sticklebacks (Rowland, 1994), in addition to being an important mate choice cue for females (Milinski & Bakker, 1990, reviewed by Rowland, 1994). Sticklebacks’ eyes have four cone pigments with visual peak absorption maximums around 360 nm (ultra-violet sensitive), 445 nm (short-wavelength sensitive), 530 nm (middle-wavelength sensitive) and 605 nm (long-wavelength sensitive) (Rowe et al., 2004, see also Lythgoe, 1979). Female sticklebacks increase sensitivity in the red spectrum during the spawning period (Cronly-Dillon & Sharma, 1968). Male three- spined sticklebacks also courted females more when illuminated by full-spectrum light including ultra-violet, compared to females presented in light lacking ultra-violet light (Rick & Bakker, 2008a). Especially the long wavelengths (“red” light) and the short wavelengths (ultra-violet) seem to be important when females courted male three-spined sticklebacks (Rick & Bakker, 2008b).

Despite the extensive scientific literature, studies on female three-spined sticklebacks with ornaments, especially red ornaments, are few. In a general overview of fishes in Maine (U.S.), several species of Gasterosteidae, including three- spined sticklebacks, were described to have red colours (Bigelow & Schroeder, 1953). In three-spined sticklebacks (sex not specified) “…the fin membrans often are red”, whereas in female three spine sticklebacks “…the whole body except the top of the back may then be reddish…” during the spawning season (Bigelow & Schroeder, 1953). In an overview of fishes in western Europe, the red colour in female three-spined sticklebacks was not mentioned whereas males were described as having red throats during the spawning period (Pethon, 1985). More recent and detailed studies reported red pelvic spines in both sexes of a population of the brook stickleback (Culaea inconstans) from Washington (Hodgson, Black & Hull, 2013, see also McLennan, 1995). Some gravid female three-spined sticklebacks from a population in Long Islands have vertical barring on the upper half of the body (Rowland, Baube & Horan, 1991; Rowland, 1994). Pelvic spines are part of the defensive armour protecting three- spined sticklebacks from predators (Moodie, 1972), which has been studied in numerous populations in North America (e.g., Moodie, 1972; Hagen & Gilbertson, 1972; Rowland, 1994), and Europe (Klepaker & Østbye, 2008; Gross, 1978). Red colour was observed at the throat of female three-spined sticklebacks from California (Pescadero Creek, von Hippel, 1999), as well as at the throat and at the membrane of the pelvic spines of females from two stream-resident three-spined stickleback populations from British Columbia (Little Campbell River, McKinnon et al., 2000). Moreover, female three-spined sticklebacks from another site in California (Matadero Creek) were reported to have red ornaments both at their throat and at their pelvic spines (Yong et al., 2013). McKinnon et al. (2000), referring to personal communications with colleagues who have observed red ornaments, wrote that such ornaments “…occur at least occasionally in other populations” of three-spined sticklebacks. Yet, only one population from Europe has been reported to have red ornamented females (Lake Nedre Vollvatn, Nordeide, 2002; Nordeide, Rudolfsen & Egeland, 2006). Females in this population were reported to have a red membrane attached to the pelvic spines but not red throats, whereas the males had both red pelvic spines and red throats. Extensive studies are absent on the occurrence of pelvic spine ornaments in male and female sticklebacks, and on carotenoid-based throat ornamentation in female stickleback populations. The few published studies have reported males to have more exaggerated red ornaments compared to females, and body lengths to be associated with the elaboration of the ornament (McKinnon et al., 2000; Yong et al., 2013). The difference in the intensity of the ornament between ovulating and non-ovulating females seems to be minor (McKinnon et al., 2000; Yong et al., 2013). Ambiguous results were reported on the relationship between the elaboration of red ornaments and body condition of sticklebacks (Hodgson, Black & Hull, 2013; Yong et al., 2013), whereas red ornaments were negatively affected by the parasitic cestoda Schistocephalus solidus (Milinski & Bakker, 1990; Barber, 2007; Candolin & Voigt, 2001; Folstad et al., 1994).

A large environmental component is expected in carotenoid-based ornaments, since animals cannot synthesize carotenoids and must acquire them through the feed (Goodwin, 1984). Empirical estimates of the relative roles of genes and environment on ornaments are contradictory, although the genetic contribution is often low (Pagani-Núñez et al., 2014; Evans & Sheldon, 2012; Hadfield & Owens, 2006; Hadfield et al., 2006; Hill, 1993a). On the other hand, some studies have shown a genetic component in carotenoid-based characters, like the red ornamented throat in male three-spined sticklebacks (Bakker, 1993), flesh colour in Chinook salmon (Oncorhynchus tshawytscha) (h2 > 0.71) (Withler, 1986) and in Arctic charr (Salvelinus alpinus) (h2 = 0.26 ± S.E. 0.16) (Elvingson & Nilsson, 1994).

The aim of this study was to give the first extensive overview of prevalence of red pelvic spine ornaments of three spine sticklebacks, from north-west European populations. Additionally, we aimed to test for potential effects of sex, body size, parasitism and season on the elaboration of the ornament. Finally, we report from an experiment where wild sticklebacks from one of the populations were crossed and their offspring reared, in order to estimate a potential genetic component of red pelvic spine ornaments in sticklebacks.

Materials and Methods

Field study

Three-spined sticklebacks were sampled at 17 sites in north-west Europe from 22 May to 20 August 2012, to estimate (i) the occurrence of individuals with red pelvic spines, (ii) how the intensity of red varied between stickleback populations, and (iii) whether the intensity of red was affected by sex, parasitism and body size. The sampling strategy was a compromise between limited financial resources available on one hand and an intension to cover as large parts of north-west Europe as possible on the other hand. Stations 8–15 (Fig. 1) were chosen due to their relatively close location to the University of Nordland and due to our prior knowledge about stickleback occurrence. The remaining stations were chosen based on information from kind colleagues, friends and relatives, on the occurrence of sticklebacks. All samples were from landlocked freshwater populations except two (no 13 and 14) which were brackish. Ten of the populations were from North Norway, one (no 7) was from mid-Norway, and the remaining six populations (no 1–6) were from the southern parts of Norway (Fig. 1 and Table 1). The altitude of the sampling sites varied from 1 to 150 m (Table 1). The majority of the samples were collected in May–July, whereas two (no 7 and 16) were sampled in August (Table 1). An additional sample was included from one (Lake Pallvann, Table 1) of these 17 sites, to examine potential change in intensity of red at the pelvic spines within the spawning season compared to 3–4 months later.

Figure 1 Map of the sampling sites.

Locations of sites where the sticklebacks were collected. Numbers correspond to “No” in Table 1.

Table 1 Information about the samping sites.

Site locations number (No), date of capture, whether sampling in freshwater (F) or marine (M) habitat, longitude/latitude, and altitude for the studied sticklebacks. The populations are numbered from south to north. One of the populations was sampled twice, first during the spawning season in May (“No 9s”), then in 3–4 months after the end of the spawning season (“No 9a”).

No	Site	Date of capture	M/F	Longitude, Latitude	Altitude (m)	
1	Øvre Sundbydam	5–6/6–12	F	59°38,492′N,10°35,239′E	50	
2	Engenvannet	14/6–12	F	59°53,754′N, 10°31,880′E	1	
3	Vassnesvannet	12/6–12	F	60°03,664′N, 05°22,206′E	4	
4	Kvernavannet	11/6–12	F	60°10,006′N, 05°23,696′E	4	
5	Myrdalsvannet	14/6–12	F	60°31,423′N, 05°39,859′E	74	
6	Nygårdsparken	14/6–12	F	60°38,228′N, 05°32,940′E	2	
7	Jonsvannet	19–20/8–12	F	63°21,941′N,10°34,879′E	150	
8	Torghatten	12/7–12	F	65°43,08′N, 12°11,282′E	3	
9s	Pallvannet	23/5–12	F	67°18,259′N, 14°24,471′E	138	
9a	Pallvannet	7/10–12	F	67°18,259′N, 14°24,471′E	138	
10	Nedre Vollvann	8–9/6–12	F	67°18,055′N, 14°26,767′E	123	
11	Frosktjønna	22/5–12	F	67°18,760′N, 14°37,658′E	70	
12	Vatnvatnet	17–20/6–12	F	67°20,030′N, 14°46,460′E	8	
13	Nord Valen	26–29/6–12	M	67°25,298′N, 13°54,202′E	1	
14	Kalven	23–28/6–12	M	67°25,298′N, 13°54,202′E	1	
15	Åsvannet	8–10/7–12	F	68°17,215′N, 16°41,027′E	79	
16	Håkøybotn	8/8–12	F	69°37,658′N, 18°43,968′E	2	
17	Skarsfjord	31/7–12	F	69°56,769′N, 18°51,621′E	3	

All fish were caught by traps. The majority of the samples were collected using traps made by cutting 1.5 l transparent soda bottles into two parts, turning the upper part (about 1/3 of the bottle) upside down, and assembling the two parts by twine. Fish from lakes 3–6 were caught by passive traps made of plexiglas (Breder, 1960), and by minnow-traps made of small-meshed nets of nylon. The traps fished during a period of 20–24 h. The sticklebacks were killed by an overdose of tricaine methanesulfonate (MS222) immediately after the traps were emptied. After the required exposure time (approximately 1–2 min), the dead sticklebacks were quickly rinsed in freshwater to remove any anaesthetic residue and placed on ice in a dark container. The sticklebacks were kept in a freezer until transported to the University of Nordland in Bodø, where they were kept in complete darkness in a −40 °C freezer until they were photographed (see below). This was to ensure that the carotenoids on the pelvic spines did not oxidize due to light exposure.

Rearing experiment

The rearing experiment lasted from June 2008 to June 2009. Parents were caught 3–16 June 2008 in Lake Nedre Vollvatn (Table 1 and Fig. 1). These fish were kept in the holding tanks for 4–7 days before sacrificed by an overdose MS222. To fertilize eggs of parents and rear their sibling groups of offspring we used the method described by Barber & Arnott (2000), with some modifications. The female’s (mother’s) abdomen was gently squeezed and eggs and ovarian fluid were collected in a Petri dish. The male’s (father’s) gonad was first removed then cut into small pieces in the Petri dish before the eggs and semen were physically mixed and left for fertilization the next 5 min. In the hatchery, the offspring in each sibling group, produced from each of the fertilizations trials, were reared separately in plastic boxes with about 160 ml volume and a continuous flow of water, as described by Rudolfsen et al. (2005). This rearing method removed potential paternal effects on the offspring. When most of the eggs in a particular sibling group hatched, the eggs and larvae were moved to a 2 l tank and for the next 3–4 weeks first fed Artemia nauplii and commercial dry feed (TetraMinBaby; Tetra, Melle, Germany), and later dry feed and chopped Chironomidae larvae. The fry sibling groups were moved to larger 7.5 l tanks 27 August, and from September–January given only Chironomidae larvae as food. In three of the 15 sibling groups approximately half of the specimens were 6 January moved to another and identical tank to avoid too high density of fish (offspring from the same sibling-group were kept in two identical 7.5 l tanks). From 26 January to 16 April the offspring in the 18 tanks (15 sibling groups) were fed dry feed containing carotenoids (astaxantin and β, β-carotene, see Appendix S1 for a recipe). The specimens in each sibling groups were fed in excess, starting with three times a day the first 3–4 weeks from start feeding, and ending with once a day the last 6 months. Excess food was always available in the tank the entire 12 months period. From October onwards, the sticklebacks experienced the natural light regime in Bodø. This feeding of the reared offspring with carotenoids resulted in intensity of red of daughters and sons overlapping to a large degree with their wild caught parents, although the offspring were slightly more ornamented than their wild caught fathers’ and mothers’ (Appendix S2). Thirty-one offspring (10.6%) from the 15 sibling groups died from the start of feeding with carotenoids 26 January to the experiment terminated 512 months later.

A potential genetic component of the intensity of red was estimated both by General Linear Models (GLM, see below) and as heritability (h2). The latter, with standard error, was estimated by the classical parent–offspring regression method for the 15 fullsib groups and 301 offspring (after discarding non-mature offspring), and adjusted for unequal family-sizes as first suggested by Kempthorne & Tandon (1953). A potential sex effect of the intensity of red (IR, see below) was handled by doing an overall regression between mother and the mean of daughters and likewise with the fathers and sons. The mean weight in each offspring group was used as a predictor in the model when estimating the regression coefficient.

Common for the field study and the rearing experiment

To quantify intensity of red colour of the spine, we took close up photos of the ventral part of the fish with raised spines including the red carotenoid-based skinfold at the basis of the pelvic spine. A standard red cardboard (227N, 1103 0964-Y-23R, Jotun A/S, Sandefjord, Norway) was included in all photos. We used a Nikon D2X digital camera (Nikon, Tokyo, Japan) with a Nikon ED AF Micro Nikkor 200 mm 1:D lens and Nikon Speedlight SB-80DX flash. The digital photos were analysed by Adobe Photoshop SC3 (San Jose, California, USA) in Red-Green-Blue (RGB) modus. We started by drawing a line in Adobe Photoshop enclosing an area around each of the right and left pelvic spine. The average density values for all three primary colours R, G, B (red, green, and blue) were quantified from all the pixels enclosed by this area for each of the two spines (Villafuerte & Negro, 1998). This was repeated for the standard cardboard in the photo. The mean R, G and B values of each of the two pelvic spines were used in further calculations. The intensity of the red colour (IR) of the skin folds of both the two pelvic spines and the red cardboard was calculated according to the formulae IR=R/R+G+B.

After calculating the IR from both the left and right pelvic spines for the first 266 fish, we estimated the coefficient of correlation between the left and right spines as RS = 0.87 (P < 0.001, N = 266, Spearman’s). Based on this relatively high correlation coefficient, we decided to measure the right pelvic spine solely for the remaining fish. We adjusted the final IR-value of the skin fold of the pelvic spines of each fish according to the IR-value of the cardboard in each photo relative to the average IR-value of all photos. Similar methods to quantify colouration have previously been applied by several authors who discussed this method of quantifying colour in ornaments, and gave more details and estimates of repeatability (Yong et al., 2013; Nordeide, Rudolfsen & Egeland, 2006; Villafuerte & Negro, 1998; Nordeide et al., 2008; Skarstein & Folstad, 1996; Skarstein, Folstad & Rønning, 2005; Neff et al., 2008). An alternative method to quantify colour, spectrophotometry, was discarded because of the small size, difficult accessibility of the ornament, and (in some individuals) un-even distribution of the colour at different parts of the spine (Fig. 2) would impede the IR-estimates. Red coloration is caused by pteridines in some fishes (Grether, Hudon & Endler, 2001). Pteridines have similar spectral properties as carotenoids, but pteridines are not extracted in acetone contrary to carotenoids (Grether, Hudon & Endler, 2001). The spines became colourless after extracting carotenoids by acetone (ES Egeland, 2005, unpublished data), which means that the red colour at the pelvic spines is caused by carotenoids and not pteridines.

Figure 2 Photos of a pelvic spine from three different sticklebacks.

Photo of the right pelvic spine from three different three-spined sticklebacks from site number 10 (see Table 1). An elaborately ornamented male with intensity of red (IR = 0.46) is shown in (A), an intermediately ornamented female (IR = 0.37) is shown in (B), and a drab female (IR = 0.35) is shown in (C). Photos by Jarle Tryti Nordeide.

All fish were measured for total length (nearest mm) and total wet weight (nearest 0.001 g) and gonads were examined for sex and whether or not they were sexually mature. All fish were visually examined externally for the microsporidian Glugea anomala and in the body cavity for potential specimens of the tapeworm S. solidus.

Several people were involved in sampling the sticklebacks at the 17 different sites, and the time from death of the fish until they were frozen differed between sites. We carried out a small experiment to test if this time difference could affect our estimates of IR. Thirty sticklebacks were captured from Lake Pallvannet (Table 1, site 9) May 16 2012, transported alive to the University of Nordland, and kept in a 300 L tank of freshwater containing hides made of plastic tubes to reduce the stress level. The fish were not fed. On May 18 the fish were killed by MS-222, and photographed three times: immediately after death (0 h), after one hour (1 h), and after four hours (4 h). The fish were held in a dark room at 4 °C between photographs. Quantifying IR-values (as explained above) revealed a mean (±S.D.) of 0.414 (±0.0318) just after death (0 h), 0.423 (±0.0238) at 1 h, and 0.414 (±0.0200) four hours after death. The difference in IR during the time interval from 0 to 4 h after death was non-significant (paired t-test: t < 0.001, P > 0.99, d.f. = 29).

The red ornament at the throat of the male sticklebacks clearly faded and nearly disappeared during handling and transportation as judged by the eye, during a period of 30–60 min (JT Nordeide, 2012, unpublished data). This observation concurs with a report by Frischknecht (1993). Contrary, the red colour of the pelvic spine ornament was much more stable and apparently not affected during the transport and handling (JT Nordeide, 2012, unpublished data).

Statistical analyses were carried out by General Linear Models (GLM) in SPSS version 20.0 (SPSS Inc., Chicago, Illinois, USA) according to Grafen & Hails (2002). None of the variables needed to be transformed to meet the assumptions of independence, heterogeneity of variance, normality of error, and linearity (Grafen & Hails, 2002).

This study was carried out in accordance with ethical guidelines stated by the Norwegian Ministry of Agriculture through the Animal Welfare Act.

Results

Field study

Photos of the carotenoid-based pelvic spine ornament of a relatively ornamented fish (IR = 0.46), and a relatively drab fish (IR = 0.35) are shown in Figs. 2A and 2C, respectively. A third photo (Fig. 2B) demonstrates the spine coloration of a moderately ornamented (IR = 0.37) but still clearly red (or reddish) fish. Individuals with carotenoid-based ornamented pelvic spines with an intensity of red (IR) ≥ 0.37 were found in all the 17 examined populations (Fig. 3). The median IR of both males and females was higher than 0.37 for both sexes in all sites except three (no 3, 4, 7), whereas in another two sites (no 11 and 15) only males (not females) had median IR ≥ 0.37 (Fig. 3). The most elaborately ornamented individuals were found in population no 10 and no 14, where a few individuals had IR > 0.50 (Fig. 3). “Sex” of the fish (entered the model as a “fixed factor”) had the strongest effect on IR in a linear mixed model (GLM) including sticklebacks from all the 17 populations (F = 101.417, Table 2). Males had a higher IR as compared to females (Figs. 3 and 4). “Population” (entered the model as a random factor) had a strong effect on IR as well (F = 64.114, Table 2 and Fig. 3). Body “length” of the sticklebacks (entered the model as a covariate) had an effect of IR (Table 2) and this association was positive (Fig. 4 and Table 2). Finally, sticklebacks infected by the parasite S. solidus had a lower IR than non-infected fish (Table 2 and Appendix S3). S. solidus were found in seven of the populations (no 6, 7, 8, 10, 12, 15 and 17) and one population (no 4) was infected by G. anomala. Running the model again including all 17 populations but excluding all parasitized fish, or only including (all) fish from the seven parasitized population only, both lead to only minor changes in the estimated parameters.

Figure 3 Intensity of red (IR) at the pelvic spines of three-spined sticklebacks from 17 different sites.

Box-Whiskers plot of intensity of red (IR) at the pelvic spines of three spine stickleback males (open bars) and females (hatched bars), from 17 different sites. “Site number” refers to Table 1. The numbers in the figure show sample size. The three dotted horizontal lines are at IR-values 0.46, 0.37 and 0.35, which represent IR-values of the pelvic spines from the three fish shown in Fig. 2.

Figure 4 Intensity of red at the pelvic spines to body length.

Scatter-plot of intensity of red (IR) at the pelvic spines plotted to body length of individual three-spined sticklebacks. Included were individuals from all 17 populations where sex could be categorized by inspecting their gonads. Open and filled circles show males and females respectively. The upper and lower lines are the regression line for males and females respectively. Note that some circles overlap.

Table 2 Testing if intensity of red at the pelvic spines is affected by the predictors.

Test statistics from a GLM type III (adjusted) sums of squares (SS) with “Intensity of red (IR)” as the response variable, and the predictors “sex” and whether or not the fish were “parasitized” (by Schistocephalus solidus) as fixed factors, “population” as random factor, and “length” as covariate. Included were sticklebacks from all the 17 different populations where sex could be categorized by inspecting their gonads.

Source	SS	d.f.	F	P-value	
Analysis of variance	
Sex	0.062	1	101.417	<0.001	
Parasitized	0.002	1	4.035	0.045	
Population	0.629	16	64.114	<0.001	
Length	0.004	1	6.381	0.012	
Term	Coeff.	SE Coeff.	t	P-value	
Coefficients	
Constant	0.3847	0.00933	41.51	<0.001	
Sex*	0.0210	0.00206	10.07	<0.001	
Parasitized**	−0.0093	0.00463	−2.01	0.045	
Length	0.00418	0.00166	2.52	0.012	
Notes.

* Females were coded as “1” and males as “2” before running the model.

** Non-infected fish were coded “0” and infected fish “1” before running the model.

No correlation was found between mean intensity of red at each of the 17 sampling sites and either the longitude of the sites (rS = 0.313, P = 0.221, N = 17, Spearman’s correlation coefficient), or mean intensity or red and altitude at each site (rS = − 0.142, P = 0.588, N = 17).

As opposed to red pelvic spines, no sign of red colour at the throat was observed in any female from any of the 17 populations. Most males in all populations sampled during the spawning season (May–June/July) had at least some red colouration at their throat, although we did not quantify male throat colour due to its volatile nature (Frischknecht, 1993).

To test for seasonality of the ornament, three-spined sticklebacks from one of the lakes (Lake Pallvann) were sampled both in the spawning season (May 2012) and 3–4 months after the end of spawning (October 2012, see Table 1). Including data from only these two samples from Lake Pallvann showed that both “Season” and “Sex” (as fixed factors) were significant, and they explained 30.1% of the variation of IR in a General Linear Model (Table 3 and Fig. 5). The effect size of the IR between the seasons was moderate especially for the females (Fig. 5), but still significant between seasons when the model was run again with only females included (F1,59 = 7.764, P = 0.007). Re-running the same model (as in Table 3) and adding body length of the fish as a covariate, gave a non-significant effect of length (F1,104 = 2.763, P = 0.099) and only minor changes for the other predictors (“Season” and “Sex”).

Figure 5 Intensity of red of the pelvic spines of sticklebacks from Lake Pallvann.

Box-Whisker plot showing median intensity of red (IR) of the pelvic spine of female and male three-spined sticklebacks from Lake Pallvann in northern Norway (Table 1 and Fig. 1). The fish were caught during the spawning season (“Spring”) and 3–4 months after the end of spawning (“Autumn”). The “Spring” and “Autumn” samples refer to “No 9s” and “No 9a” in Table 1, respectively.

Table 3 Testing for the effect of season and sex on redness of the pelvic spines.

Testing for effect of season and sex on intensity of red (IR) on the pelvic spines of three-spined sticklebacks. The table shows test statistics from a GLM type III (adjusted) sums of squares (SS) with “Intensity of red (IR)” as the response variable, and the predictors “sex” and “season” as fixed factors. Included are fish from Lake Pallvannet which could be categorized by sex by inspecting their gonads. “Season” is whether the fish were sampled during the spawning season (23 May) or 3–4 months after the end of the spawning season (7 October). The model explained 30.1% of the variation in the data (Adjusted R2 = 0.301).

Source	SS	d.f.	F	P-value	
Analysis of variance	
Sex	0.012	1	20.864	<0.001	
Season	0.012	1	21.702	<0.001	
Error	0.060	105			
Total	17.695	108			

We then tested for association between condition of the fish and IR of the pelvic spines of the three-spined sticklebacks from the October-sample from the same lake (Lake Pallvannet, no 9a in Table 1). Condition was estimated as the residuals of fish weight (g) on length cm in a GLM, (weight needed no ln-transformation to be linearly associated with length). None of the predictors “Residuals of weight on length” (as explained above) as a covariate (F1,43 = 2.555, P = 0.117) or “Sex” as a fixed factor (F1,43 = 2.882, P = 0.097) was associated with the response variable IR of the pelvic spines. Running the same model again after including (body) “Length” (as a covariate) as a third predictor did not explain a significant part of the variation in the response variable either (F1,42 = 0.064, P = 0.802).

Rearing experiment

No association was found between the mean intensity of red spines in each of the tanks with offspring groups (as the response variable), and the mean weight gain during the period when fed carotenoids which started 26 January and ended 16 April, as covariate (F1,17 = 0.034, P = 0.855, R2 = − 0.060). This indicates that the intensity of red of the specimens in each of the 18 tanks in total (15 sibling groups) was not a direct effect of the amount of carotenoid-feed consumed during this period.

In a GLM with “offsprings’ IR” as the response variable, and the predictors “sex,” “length,” “mother’s IR,” and “father’s IR” as covariates, only the latter predictor (“father’s IR”) was non-significant (see Appendix S4). This model explained 31.3% of the total variance. Males had highest “offsprings’ IR” values (residuals after adjusting for “length”) (mean ± S.E.: 0.36 ± 0.099), followed by immatures (−0.19 ± 0.146), and females (−0.203 ± 0.076). When running the same model after excluding immature offspring, the predictor “sex” was still significant (p < 0.001). Based on these results, we decided to analyse male and female offspring in separate models.

Starting with female offspring, “offspring IR” was associated with “length” and with the interaction term between the two other covariates “father’s IR” and “mother’s IR” (Table 4). The p-values of “father’s IR” and “mother’s IR” were less than 0.031. The model explained 25.1% of the variation, with “length” explaining 21.1% and the remaining (“genetic”) terms (“father’s IR”, “mother’s IR”, and “father’s IR” x “mother’s IR”) explaining 4.0% of the variance in the model. The residuals of daughters’IR (adjusted for body length) plotted against their mother’s IR, is shown in Fig. 6A.

Figure 6 Association between ornaments in mothers and their offspring.

Scatter plot of the intensity of the red pelvic spines (IR) of (A) mothers and daughters and (B) mothers and sons, and their linear regression lines. Y-axis shows residuals of IR after adjusting for length of the (A) daughters and (B) sons.

Table 4 Intensity of red at the pelvic spines from the rearing experiment.

Daughters from the rearing experiment: Test statistics from a GLM type III (adjusted) sums of squares (SS) with intensity of red colour of the offsprings’ pelvic spines (IR) adjusted for body length as the response variable. Predictors are “length,” and IR of father’s pelvic spines (“Fathers’ IR”) and IR of mothers pelvic spines (“Mothers’ IR”), and these were entered the model as covariates. Adjusted R2 = 0.251.

Source	SS	d.f.	F	P-value	
Analysis of variance	
Length	0.006	1	19.210	<0.001	
Father’s IR	0.001	1	4.786	0.030	
Mother’s IR	0.002	1	5.175	0.025	
Mother’s IR* Father’s IR	0.001	1	4.801	0.030	
Error	0.038	129			
Total	21.680	134			
Term	Coeff.	SE Coef.	t	P-value	
Coefficients	
Constant	−1.693	0.893	−1.896	0.060	
Length	0.001	<0.001	4.913	<0.001	
Father’s IR	5.203	2.378	2.188	0.030	
Mother’s IR	5.546	2.438	2.275	0.025	

Concerning male offspring, residuals of offspring IR was positively associated only with “length” (F = 27.223, P = < 0.001), whereas the p-value of “mothers IR” was slightly non-significant (F = 3.202, P = 0.077) (Appendix S5). This model explained 23.1% of the variation, with “length” explaining 21.4% leaving 1.7% of the variation to be explained by the (“genetic”) term “mothers IR.” Scatter plot of residuals son’s IR (adjusted for length) against “mother’s IR” is shown in Fig. 6B. Running the model again after removing the one male offspring (“outlier”) with a very drab mother (“mother’s IR” <0.34), resulted in also “mother’s IR” becoming significant (F = 2.068, P = 0.040), and the model explained slightly more of the variance (R2 = 0.244).

Finally, including only “immature” offspring, their IR as response variable was not significantly associated with neither “length” (F1,54 = 0.432, P = 0.514), or their “mother’s IR” (F1,54 = 0.044, P = 0.835) as covariates, and the model did not explain much variation (R2 = 0.009).

The classical parent-offspring regression method (see ‘Materials and Methods’) gave an estimated heritability of 0.14 (S.E. = 0.22) when data from all mature offspring and all parents were pooled. This heritability estimate was not significantly different from zero.

Discussion

Field study

This first extensive study of prevalence of female ornaments in three spine stickleback populations suggested that carotenoid-based pelvic spine ornament is widespread among north-west European populations. Females and males with red ornamented pelvic spines were present in all the 17 populations examined, abundant in most of them, and quite conspicuous in several populations. This widespread occurrence of the red pelvic spines among populations has not been reported before, and adds another interesting aspect to the three-spined stickleback as a model species in studies of sexual selection. Individuals with red spines have so far been reported in only three North-American populations of sticklebacks (McKinnon et al., 2000; Yong et al., 2013) and one population from Europe (Nordeide, 2002; Nordeide, Rudolfsen & Egeland, 2006). The great variation in the exaggeration of carotenoid-based ornaments in the 17 stickleback populations in the present study concurs with similar studies on male and female House Finches (Carpodacus mexicanus) in North America (Hill, 1993a; Hill, 1993b). The variation between populations of House Finches was suggested to reflect local and regional variation in dietary carotenoids pigments availability (Hill, 1993a). We have no data about carotenoids availability in the 17 study cites and hence cannot confirm or dispute the importance of variation in available pigments in the present study.

Some of the results from the present study concur with previous studies on stickleback ornaments, whereas other results do not. For example, the red throat ornament (contrary to red spines) was totally absent in all females from the 17 populations and this is contrary to findings in the three abovementioned North American three-spined stickleback populations (McKinnon et al., 2000; Yong et al., 2013). We have also confirmed that the pelvic spines of males are more intensely red than the spines of the females, as reported by Yong et al. (2013) for both throat and pelvic spine ornaments, and McKinnon et al. (2000) for the throat ornament in sticklebacks. Body length of the fish was associated with IR of the pelvic spines when all 17 populations were analysed together. This result agrees with reports on female body size and throat colour by McKinnon et al. (2000) and both body size and throat colour and body size and pelvic spine colour by Yong et al. (2013). Presence and absence of the parasite S. solidus explained a significant although minor part of the variation in IR, supporting previous studies suggesting a negative association between red (throat) ornament and infection of several parasite species including S. solidus (see references in ‘Introduction’). The elaboration of the red spine ornament was higher during the spawning season compared to 3–4 months after spawning for both sexes, although many male and female sticklebacks were still red outside the spawning season (Fig. 5). Similarly, McKinnon et al. (2000) and Yong et al. (2013) reported small and non-significant differences in the throat ornament between ovulating and non-ovulating females, and they suggested that IR of females does not signal readiness to spawn (McKinnon et al., 2000; Yong et al., 2013).

In the autumn sample from Lake Pallvann we found no association between body condition of the fish and the intensity of red at their pelvic spines. Again, our result concurs with the report by Yong et al. (2013) on lack of association between red throat ornament and condition in three spine sticklebacks, and these authors suggested sexual selection to be of limited importance in the evolution of female ornaments. In female brook sticklebacks, however, Hodgson, Black & Hull (2013) reported a positive association between pelvic spine colour and condition.

We cannot rule out the possibility that lack of detailed information about start and end of the spawning season for each of the 17 populations and our sampling of populations over a several weeks long period, might have influenced our estimates of IR. Different stickleback populations were probably sampled in different parts of their spawning period and the intensity of the ornaments may vary during the spawning period. For example, fish in southern populations probably start spawning earlier than in northern populations. On the other hand, we find large variation in IR between fish from different populations located geographically close to each other at similar altitude and sampled within a few days, like for example populations 3–6 and populations 9 and 11. These observations leave little doubt that ornaments from different populations vary in exaggeration. This interpretation received support from the relatively small effect size revealed by comparing IR during and 3–4 months after the spawning season for site number 9.

Rearing experiment

The results from the rearing experiment may be interpreted as the genetic component of red pelvic spines is weak or even perhaps not different from zero. The significant association between the response variable “daugther’s IR” and each of the predictors (i) “father’s IR,” and (ii) “mother’s IR,” and (iii) the interaction between “father’s IR” and “mother’s IR,” together explain only 4% of the variation in the model. Such a weak genetic component concurs with our low heritability estimate, which was not significantly different from zero. The lack of significance in our heritability estimate was due to the large standard error which is evidently a direct function of the number of crosses and offspring (e.g., Dupont-Nivet, Vandeputte & Chevassus, 2002). A larger data set would be needed to reveal a significant estimate when heritability is this low in order to avoid Type II-errors. Such a low genetic component concurs with previous estimates in other species like plumage coloration in blue tits (Cyanistes caeruleus) (Hadfield et al., 2006) and canaries (Serinus canaria) (Muller et al., 2012). A low correlation between carotenoid-based coloration of nestlings and that of their parent great tits (Parus major) (Pagani-Núñez et al., 2014) has also been documented. On the other hand, a significant genetic component has been reported for both ornamental and non-ornamental traits in several fishes including sticklebacks (see ‘Introduction’). Fish utilize carotenoids poorly and retention of astaxanthin in the muscle of for example Atlantic salmon (Salmo salar) is less than 12% (Bjerkeng, 2008). This is partly due to poor absorption of astaxanthin from the gut (Bjerkeng, 2008). Individual variation in the efficiency to utilize carotenoids for ornamental display may be a functional explanation for the significant genetic component in carotenoid-based throat and flesh coloration in sticklebacks and salmon, and may explain individual variation in the redness of pelvic spines in sticklebacks as well.

We cannot rule out the importance of assumptions not accounted for in this experiment. Firstly, our estimate of a weak genetic component may be influenced by the different environmental conditions experienced by parents and offspring in this study. Additionally, potentially different variants of carotenoids in wild-caught parents and their reared offspring may have contributed to low heritability estimates. Our design removed potential effects of fathers on the offsprings’ ornament (see Material and Methods). However, we cannot exclude potential effects of mothers, although we do not expect large maternal effects in IR of the offspring ornament. This is based on observations of the offspring’s spines initially being very drab. The spines turned red after the offspring were given feed containing carotenoids from January onwards (see Material and Methods) (JT Nordeide, 2009, unpublished data). Finally, some of the variation in the intensity of the ornamentation of the offspring may be due to differences in the amount of food eaten, and hence amount of carotenoids consumed. On the other hand, feed containing carotenoids were available to the fish in excess at all times, and no association was found between weight gained during feeding with carotenoids and mean IR of their ornament (see Material and Methods).

To conclude, this study suggests that both male and female individuals with reddish and often conspicuously red ornamented pelvic spines, are common in north-west European three-spined stickleback populations. Males were more ornamented than females, and the fish were more ornamented during the spawning period than after spawning. The genetic component of the intensity of red spines seems to be low. This study gives little support for either red spines signalling spawning readiness, or of sexual selection being important for the evolution of the ornament. The potential adaptive function of the ornament, and how it evolved, remain largely unexplained.

Supplemental Information

Appendix S1 Recipe for feed

Recipe for feed given to stickleback offspring in the rearing experiment.

Click here for additional data file.

Appendix S2 Intensity of red of the pelvic spines of fish from the rearing experiment

Box–Whiskers plot of the intensity (IR) of the carotenoid-based ornament at the pelvic spines of the mothers (N = 15) and fathers (N = 15) in the artificial fertilizations, and their immature offspring (sex not identifiable, N = 55), daughters (N = 134) and sons (N = 103). Offspring IR was measured at an age of one year. Data are not adjusted for length.

Click here for additional data file.

Appendix S3 Intensity of red of pelvic spines of parasitized and unparasitized sticklebacks

Mean residuals of intensity of red (IR) of the pelvic spines of female (upper figure) and male (lower figure) sticklebacks, after adjusting for length of the fish. Fish infected by the tapeworm Schistocephalus solidus are shown as open circles, whereas uninfected ones are represented by filled circles. Lower and higher intervals shows 25 and 75%, respectively. Only populations with infected individuals are shown. Population numbers refer to Table 1. The numbers in the figure show number of sticklebacks examined: the first number (in italics) is number of sticklebacks infected, whereas the last number is number of fish not infected in each population.

Click here for additional data file.

Appendix S4 Test statistics from the rearing experiment

All offspring (immature/unknown, males and females) from the rearing experiment. Test statistics from a GLM type III (adjusted) sums of squares (SS) with intensity of red colour of the offsprings’ pelvic spines (IR) as the response variable, and the predictors are “Offspring’s sex” (fixed factor), and offsprings’ “length”, IR of mothers (“Mothers’ IR”) and fathers (“Fathers’ IR”) as covariates. Adjusted R2 = 0.313.

Click here for additional data file.

Appendix S5 Test statistics for sons from the rearing experiment

Sons from the rearing experiment: Test statistics from a GLM type III (adjusted) sums of squares (SS) with intensity of red colour of the offsprings’ pelvic spines (IR) as the response variable, and effects of “length” and IR of mothers pelvic spines (“Mothers’ IR”) as covariates. Adjusted R2 = 0.223.

Click here for additional data file.

Data S1 This dataset was used when preparing Table 2, Figs. 3 and 4 and Appendix S3.

Click here for additional data file.

Data S2 This data-set was used when making Table 4, Fig. 6 and Appendices S2, S4 and S5.

Click here for additional data file.

Data S3 This data-set was used when preparing Table 3 and Fig. 5.

Click here for additional data file.

Randi Restad Sjøvik taught JTN how to rear fish, Bjørnar Eggen and Sylvie Bolla provided Artemia as feed, and several colleagues, friends and family helped collecting the sticklebacks in the wild.

Additional Information and Declarations

Competing Interests

Author Contributions

Animal Ethics

The authors declare there are no competing interests.

CR Amundsen and JT Nordeide conceived and designed the experiments, performed the experiments, analyzed the data, contributed reagents/materials/analysis tools, wrote the paper, prepared figures and/or tables, reviewed drafts of the paper.

HM Gjøen analyzed the data, contributed reagents/materials/analysis tools, wrote the paper, reviewed drafts of the paper.

B Larsen analyzed the data, contributed reagents/materials/analysis tools, reviewed drafts of the paper.

ES Egeland contributed reagents/materials/analysis tools, wrote the paper, reviewed drafts of the paper.

The following information was supplied relating to ethical approvals (i.e., approving body and any reference numbers):

This study was carried out in accordance with ethical guidelines stated by the Norwegian Ministry of Agriculture through the Animal Welfare Act. According to these guidelines we are not supposed to—and therefore do not have—a specific approval or approval number.

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
