# Peer review of "Conspicuous carotenoid-based pelvic spine ornament in three-spined stickleback populations—occurrence and inheritance"

_PeerJ, doi:10.7717/peerj.872_

## Round 0.1 · original submission · Major Revisions

Please revise as requested by the reviewers.

·

Basic reporting

Regarding “Conspicuous carotenoids-based pelvic spine ornament in three-spine stickleback populations – occurrence and inheritance” by Amundsen and coworkers.

The manuscript gives the occurrence (although incomplete) of red pelvic spine coloration of populations of three-spined sticklebacks in Norway, its relationship to parasites and individual condition and its sexual dichromatism. Additionally, the manuscript includes an experiment on heritability of the trait.

The strong part of the manuscript is the documentation of the common occurrence of red pelvic spine coloration in all sampled populations and, as this is not a previously well described trait in three-spined sticklebacks, I think it merits publication. Yet, the manuscript does not clearly inform about why the particular populations where included. Have they been selected at random or have the authors tried to map a largest possible geographical range? Details about the selection procedure of the included populations should be included.

More important, as the authors also detail in the last part of the discussion (lines 385 to 418) there are good reasons to believe that the rearing experiment should give low heritability estimates. I think the manuscript would be much better if the entire rearing experiment was omitted. The heritability of spine coloration is clearly an interesting subject, as the trait seem to be much more stable than the belly coloration. Why not present results from the rearing experiment into a new manuscript where F2 generations of laboratory stocks are compared with the F1 experimental generation (i.e., where to generations experience the same environmental background). As it now stands with a wild caught parental and a F1 generation of experimental (artificially feed) individuals you run into all kinds of problems.

Additionally, the location of the population included in the study spans over a large altitude and longitude and although this information is available to the authors (see table 1) no attempt is done to see if such variables can be used as predictors of differences in spine coloration between populations. This attempt could be included.

I have lots of comments on the details of the presentation and have sent this to the authors.

Experimental design

see above

Validity of the findings

see above

Additional comments

see above

Reviewer 2 ·

Basic reporting

The English has problems in spots, and thorough proofing is recommended, also to remove minor errors.

Specific Comments:
Abstract, title: Three-spine and three spine – be consistent.
"Carotenoids-based" should be carotenoid-based

Line 34-35: citation format is not consistent
Line 36: probably more accurate to say constrain investment in offspring rather than female fitness
47: should be "explanations for"
50: should be "and evolutionary biology"
52: should be "defensive armour"
54-55: no consistent approach to order of citations
68: should be "females"
162: 6 January out of place?
202: the P value is a significance value? That does not look correct.
319: not sure why father's IR is not mentioned-please explain.
334: should be "that the caroten…"
393: I think they mean the number of crosses and offspring? Better to be more explicit.
399: perhaps this will be fixed by the journal, but et al. should be italicized throughout.

Literature cited: the formatting is not consistent, in particular the use of periods with initials.
515: typo

Experimental design

My concerns are minor but worth addressing:

192: what sort of line?
205: can they explain the adjustment more fully?
285: was any adjustment made to account for variation in female gravidity? If not, this may be the cause of the negative result, at least for females

Validity of the findings

My only concerns are minor and more appropriate to the previous section.
The interpretation of the results is generally cautious and reasonable.

Additional comments

This is an interesting study that seems to have been carried out competently and using relatively standard protocols.

---

## Round 0.2 · accepted · Accept

Thank you for your conscientious attention to detail.